# Intensity and Duration of Vibration Emissions during Shipping as Interacting Factors on the Quality of Boar Semen Extended in Beltsville Thawing Solution

**DOI:** 10.3390/ani13050952

**Published:** 2023-03-06

**Authors:** Tim Hafemeister, Paul Schulze, Christian Simmet, Markus Jung, Frank Fuchs-Kittowski, Martin Schulze

**Affiliations:** 1Institute for Reproduction of Farm Animals Schönow, 16321 Bernau, Germany; 2Environmental Computer Science, HTW Berlin-University of Applied Sciences, Wilhelminenhofstr. 75a, 12459 Berlin, Germany; 3Minitüb GmbH, 84184 Tiefenbach, Germany

**Keywords:** artificial insemination, boar semen, boar sperm quality, shipping, vibrations, duration

## Abstract

**Simple Summary:**

In the modern pig reproduction system, boar semen is a high-quality but perishable product that is distributed from only a few central boar stations to the sow farms. Like all parts of the production chain, the shipment of boar semen can affect sperm quality to varying degrees. In the present study, the loss of quality in response to increasing intensities and duration of vibration emissions was examined, and interaction of both factors was found. During normal transport conditions, vibration emissions are expected to be low, resulting in a low loss of quality. Nevertheless, the duration of transport should also be considered for the maintenance of boar semen quality at a high level.

**Abstract:**

Vibration emissions during the transport of boar semen for artificial insemination (AI) affect sperm quality. In the present study, the common influence of the following factors was investigated: vibrations (displacement index (D_i_) = 0.5 to 6.0), duration of transport (0 to 12 h) and storage time (days 1 to 4). Normospermic ejaculates were collected from 39 fertile Pietrain boars (aged 18.6 ± 4.5 months) and diluted in a one-step procedure with an isothermic (32 °C) BTS (Minitüb) extender (*n* = 546 samples). Sperm concentration was adjusted to 22 × 10^6^ sperm·mL^−1^. Extended semen (85 ± 1 mL) was filled into 95 mL QuickTip Flexitubes (Minitüb). For transport simulation on day 0, a laboratory shaker IKA MTS 4 was used. Total sperm motility (TSM) was evaluated on days 1 to 4. Thermo-resistance test (TRT), mitochondrial activity (MITO) and plasma membrane integrity (PMI) were assessed on day 4. Sperm quality dropped with increasing vibration intensity and transport duration, and the effect was enhanced by a longer storage time. A linear regression was performed using a mixed model, accounting for the boar as a random effect. The interaction between D_i_ and transport duration significantly (*p* < 0.001) explained data for TSM (−0.30 ± 0.03%), TRT (−0.39 ± 0.06%), MITO (−0.45 ± 0.06%) and PMI (−0.43 ± 0.05%). Additionally, TSM decreased by 0.66 ± 0.08% with each day of storage (*p* < 0.001). It can be concluded that boar semen extended in BTS should be transported carefully. If this is not possible or the semen doses are transported a long way, the storage time should be reduced to a minimum.

## 1. Introduction

The purpose of pig breeding farms is to produce boar semen for the artificial insemination (AI) of sows. This perishable product must endure transport over long distances while maintaining its quality [1,2]. The implementation of quality assurance programs helps to ensure the high quality of AI doses is maintained throughout production; however, the targeted monitoring of boar semen ends at the production line. There are currently no international standards for the transport of boar semen. Over the past decade, the average total sperm number per dose from 28 European AI centers has been reduced by about one-fifth, while sperm quality has steadily increased [3]. Strategies such as post-cervical or fixed-time AI [4] have contributed to a further reduction in sperm count per sow with comparable fertilization success. This has not only resulted in economic benefits but also contributed to environmental protection by reducing the number of animals needed in each AI center. For these reasons, the implementation of measures to maintain sperm quality from the point of production to insemination has become a topic of interest for several research groups. Studies have found that a number of factors have an impact on sperm quality during transport. These include transport temperature and type of semen extender [5] as well as the intensity [6] and duration [7] of exposure to vibration. Taking these previous findings into consideration, the present study aims to fill a major knowledge gap by investigating the relationship between the duration and intensity of vibration emissions to find a cut-off value for harmful vibrations.

A mobile sensing app able to measure vibrations during transport has been developed [6,8], and Schulze et al. showed the impact on boar semen quality after simulating these vibrations for six hours. This system was enhanced with an external measuring device [9] and the displacement index (D_i_) to quantify the intensity of vibration emissions within transport boxes during boar semen deliveries under everyday production conditions [10]. In the present study, these findings were simulated under laboratory conditions. In this case, the intensity of vibrations and duration of transport were not considered to be separate factors, and their interaction was examined in a wide range. For this purpose, transport times from 30 min up to 12 h, road type and speed-dependent low and high vibrations were taken into account [10]. The impact of these factors on boar semen quality was evaluated using an extended spectrum of methods with the aim of providing new and comprehensive insights into the impact of vibration emissions on the quality of boar semen. This information can then be used to develop recommendations for implementation in the field.

## 2. Materials and Methods

### 2.1. Chemicals

All chemicals used in this study were of analytical grade. Unless stated otherwise, they were purchased from Merck (Darmstadt, Germany) and Roth (Karlsruhe, Germany). Fluorescein isothiocyanate-conjugated peanut agglutinin (FITC-PNA), Pisum sativum agglutinin (FITC-PSA) and rhodamine 123 (R123) were obtained from Sigma-Aldrich (Steinheim, Germany), whereas propidium iodide (PI) was purchased from Thermo Fisher Scientific Inc. (Darmstadt, Germany).

### 2.2. Semen Processing

All AI doses examined in this study (*n* = 546) were produced in a commercial AI center in Germany. Ejaculates were randomly collected from 39 fertile Pietrain boars (aged 18.6 ± 4.5 months) using the double gloved-hand method. After quality control to ensure they fulfilled minimum requirements for commercial use in AI, the semen was diluted in a one-step procedure with an isothermic (32 °C) Beltsville Thawing Solution (BTS) extender (Minitüb GmbH, Tiefenbach, Germany). Semen was placed in 95 mL QuickTip Flexitubes^®^ (Minitüb GmbH) with a sperm concentration adjusted to 22 × 10^6^ sperm·mL^−1^ and a filling volume of 85 ± 1 mL. Compliance with the minimum requirements of the German Livestock Associations standard for short-term extender (total sperm motility ≥ 70%, morphologically abnormal sperm ≤ 25%, sperm with cytoplasmic droplets ≤ 15%) was confirmed at the reference laboratory for spermatology at the Institute for Reproduction of Farm Animals Schönow (IFN).

### 2.3. Simulation of Transport Vibration Emissions

The experimental design is based on an analysis of the transport conditions during the shipment of boar semen according to Hafemeister et al. [10]. An orbital shaker IKA MTS 4 (Laborgeräte München, Germany) with circular horizontal shaking movements was found to be able to best simulate the vibrations from the field. However, when comparing different devices of the same model, different D_i_ values were found, although the same settings were used. In order to eliminate this possible source of error, the system was modified to directly display the real-time D_i_ values generated by the vibrations.

The AI doses were placed in a polystyrene box equipped with the mobile measuring device. To simulate otherwise optimal transport conditions, the room in which the orbital shaker was kept was dark and maintained at 17 °C. The box with the AI doses was strapped to the orbital shaker, and the displacement index (D_i_ ± 0.1) was set by the rotation speed of the shaker. One sample of each ejaculate was removed from the shaker every hour over a simulation period of 12 h. In order to simulate short distances, an additional sample was removed after 0.5 h of treatment. Thus, together with the unshaken control sample (duration = 0 h), 14 AI doses were analyzed per ejaculate. Furthermore, three ejaculates could be treated simultaneously (*n* = 42) for each vibration intensity. Slight deviations in the desired displacement index were caused by weight changes as a result of the hourly removal of samples; this was identified through real-time monitoring and corrected accordingly. After treatment and in accordance with best practices, all samples were stored motionless in a dark temperature-controlled cabinet (17 °C) while awaiting further investigation. The day of treatment was defined as day 0 (d0).

To obtain an overview of the response of different vibration intensities on sperm quality, only total sperm motility was determined in a first preliminary round of experiments (*n* = 210 samples, Table 1). In the main round, the samples were examined using an extended spectrum of methods (*n* = 336), which are described below.

### 2.4. Assessing Sperm Quality

#### 2.4.1. Total Sperm Motility

After careful resuspension, an aliquot of 1.5 mL was incubated for 10 min at 38 °C, and total sperm motility (TSM) was assessed daily on day 1 (d1) through day 4 (d4) using the computer-assisted semen analysis (CASA) system AndroVision^®^ (Minitüb, Germany). An aliquot of 3.0 µL was placed on a preheated four-chamber slide (Leja products B.V., Nieuw-Vennep, The Netherlands), and at least 1000 sperm cells were analyzed. Sperm were defined as motile when showing an amplitude of lateral head displacement (ALH) > 1.0 µm and a velocity curved line (VCL) > 24.0 µm × s^−1^.

Additionally, on d4, a thermo-resistance test (TRT) was performed. For this purpose, a 10 mL sample was incubated for 300 min at 38 °C with exposure to air; afterwards, the motility was determined as described above.

#### 2.4.2. Mitochondrial Activity and Plasma Membrane/Acrosome Integrity

To examine molecular processes in sperm cells, two flow cytometric assays were performed on d4. In a double staining with R123/PI, the percentage of viable spermatozoa with active mitochondria (MITO) was recorded (R123 pos., PI neg.), as described previously [11]. A triple staining technique using PI, PNA and PSA was performed to determine the plasma membrane and acrosome integrity (PMI) of the spermatozoa according to Schulze et al. [12]. PMI reports the percentage of spermatozoa with intact plasma- and acrosomal membranes (PI neg., PNA neg. and PSA neg.).

### 2.5. Data Processing and Statistical Analysis

Data processing and statistical analysis were performed with R [13], extended by the packages *lmerTest* [14] for linear mixed models and *ggplot2* [15] for graphical representation.

To compensate for the slightly divergent initial qualities of the individual ejaculates, the difference between each treatment sample and the unshaken control sample on the same examination day was calculated using Equation (1):(1)$$\Delta \left[\mathrm{Parameter}\right]={\left[\mathrm{Parameter}\right]}_{\mathrm{Treatment}}-{\left[\mathrm{Parameter}\right]}_{\mathrm{Control}}$$

The assignment of treatment and control parameters results in negative values representing a decreasing parameter due to the treatment. The results for each combination of D_i_ and duration are reported as mean with standard deviation (mean ± SD) and mean difference with standard error (mean-Δ ± SE), respectively.

To investigate the relationship between D_i_, duration and sperm quality characteristics, linear regression was performed. Based on the assumption that the ejaculates of different boars respond differently to vibration emissions, a mixed model was calculated that included the boar as a random effect. The interaction of D_i_ and duration was used as a fixed effect explaining the measured sperm quality characteristics. For total sperm motility, the examination day (d1–4) was included in the model. This represents different storage times after shipping in order to reflect the influence of transport on storability of semen. Several model variants with different combinations of predictors were tested. A *p*-value < 0.05 was considered a significant predictor, and only significant predictors were used to explain the data. For further selection, the model variants were compared in an ANOVA, and the model with the lowest *Akaike information criterion* (AIC) was selected.

## 3. Results

Within the following 13 experimental weeks, 546 AI doses from 39 different boars were treated and observed with 8 different vibration intensities (Table 1). The TSM of the ejaculates at d0 was determined from the unshaken control samples and averaged 79.4 ± 6.2% (mean ± SD), with morphological abnormalities < 20%. To obtain an overview of all subgroups formed by the combination of simulation duration and D_i_, only the data from three different durations are summarized in Table 2, representing short (1 h), middle (6 h) and long (12 h) transport conditions. The detailed results of all durations can be found in Appendix A.

In order to comprehensively describe the relationship of the measured values, a mixed linear regression model was constructed for each sperm quality parameter. The predictors D_i_ and duration applied individually did not show a significant impact on the data (*p* > 0.05). Only the storage time for TSM and the interaction term D_i_ × duration had a significant effect (*p* < 0.001) on the data and were used as fixed effects. The different boars were included as a random effect on the intercept and slope. The correlation between fixed and random effects is low (|r| ≤ 0.32), which justifies the use of the mixed model. The regression analysis showed that with each increase of the product (D_i_ × duration) by 1, sperm quality parameters decreased by 0.30–0.45% (Table 3). The TSM additionally decreased by 0.65% with every further storage day. Figure 1a shows the summarized data of TSM on d4, including the mean absolute and mean difference values (mean-Δ TSM) as well as the regression line of the calculated mixed model. Similarly, the data for TRT, MITO and PMI are shown in Figure 1b and Figure 2a,b, respectively.

## 4. Discussion

The aim of this study was to determine the influence of different vibration intensities during transport on BTS-extended boar semen. The results demonstrate a significant negative impact caused by the interaction of the intensity and duration of vibration emissions. Our measurements also show that even low vibration intensities impair sperm quality if AI doses are subjected to them for a long time. Setting a cut-off value for harmful vibrations, as originally intended by this study, is, therefore, not very useful. Rather, it should generally be considered important to minimize the time BTS-extended boar semen is exposed to even low-vibration intensities during transport.

As of 2023, genetically valuable boars are housed in a few AI centers. Their diluted ejaculates are shipped to sow farms across long distances using a wide variety of vehicles. Transport times up of to 12 h and distances up to 600 [2] and 1500 km [10] or more are reported. As a negative influence of rotation of AI doses during storage on sperm quality has been previously demonstrated [16], vibrations during the transport of boar semen have become a relevant topic and the newfound focus of research groups. Schulze et al. conducted an experiment in which vibrations within a vehicle, caused by varying road conditions, were detected by a mobile sensing app. They were then able to simulate these findings in a laboratory setting by using a shaker and adjusting the rotation speeds (revolutions per minute, rpm) in the laboratory [6]. Later studies used these rpm values as a reference for their experimental designs [5,7,17]. Although, until now, only Tamanini et al. have adjusted rpm for a different shaker model through calculations, the comparison of the rotation speeds is limited because varying loading affects the resulting vibrations [18]. Furthermore, in the previous experiments, only two shaking frequencies with one duration [6], one frequency with varying durations [7] or only one duration were studied [5,17], and other factors were added to the experimental designs. A comprehensive consideration of multiple intensities with different exposure times has been lacking, and our study aimed to elucidate this gap in knowledge.

Total sperm motility is the parameter most commonly used in AI centers to assess the quality of ejaculates [19]. Although high values do not guarantee high fertility capacity, low values are indeed associated with small litter sizes [20]. Figure 1 shows that total sperm motility in the AI doses decreases the longer vibrations are applied, and the effect becomes greater the stronger the vibrations are. Regression analysis suggests that short heavy vibrations affect TSM in the same way as long mild ones. This confirms the results from the studies that have separately examined different levels of intensity or duration. Although in Schulze et al. [6] the comparison between mild vibrations (100 rpm) and the control group was not significant, a trend could already be identified. Thus, it can be postulated that a shaking time longer than 6 h would result in a significant decrease in total sperm motility.

Tamanini et al. [7], who simulated mild vibration emissions, were able to show a linear relationship between the interaction of duration and extender on motility loss, but the reduction in short-term extender (BTS) was not significant. In contrast, our regression model also explains low motility losses, which arise from mild vibrations. In addition, although the measured values show that motility decreases with each storage day, no relationship with the vibration treatment could be determined with the regression model. Therefore, it can be assumed that the observed slight decrease in total sperm motility corresponds to the normal loss rate during storage [21]. Slight variations in the fertilization capacity of an ejaculate, which may remain hidden from common motility assessment, can be revealed through the application of a thermo-resistance test [22]. Overall, the results show the same pattern as for TSM, but the drop in the values calculated by the regression model does not behave as assumed, as the regression lines at D_i_ = 4.0 and 5.0 are flatter than expected. This could be explained by the boar-specific effect [23], which is influenced by the individual composition of the seminal plasma and the sperm cell membranes [24]. It has been demonstrated that seminal plasma influences sperm cryotolerance [25]. Thus, variable tolerance to vibration emissions is also conceivable and should be the subject of further research.

A limitation of this study is the use of BTS semen extender. BTS is intended for use as a short-term extender; therefore, extended AI doses should be inseminated within 3–4 days. The extender contains a simple buffering system based on bicarbonate [26]. Schulze et al. [6] postulated that vibration emissions result in the loss of CO_2_, leading to an increase in pH and corresponding decrease in sperm quality [27]. A relevant change in pH was also identified in a more complex and robust buffering system by Tamanini et al. [7] when examining the long-term extender Androstar^®^ Plus. However, Paschoal et al. [5] were able to show a vibration effect without a significant pH change, suggesting that a pH change is not the only reason for an impairment in sperm quality due to vibration emissions.

Positive rheotaxis is characterized by hyperactivated motility against a fluid flow to guide sperm cells through the female genital tract [28] and has also been detected in liquid-preserved boar semen [29]. Vibration-induced movement in the fluid of the AI dose could trigger rheotactical behavior of spermatozoa and reduce the quality of the insemination portion through the early consumption of energy. The treatment-dependent decrease in mitochondrial activity would support this assumption. Although not using the exact inducing pathways, hyperactivation and capacitation of spermatozoa are closely related [30]. A capacitation process that has already occurred could explain the decreasing percentage of spermatozoa with intact plasma and acrosomal membranes. Another explanation, previously discussed by other authors [6,7], are reactive oxygen species (ROS). The damage to the cell membranes occurs due to the shearing forces of oxidative stress. Our results will help future investigations elucidate the underlying molecular mechanisms.

As our study design was intended to describe the impact of vibration and duration across a broad spectrum of these, extreme situations were also investigated, for example, through the simulation of a transport duration of 12 h. While such transport lengths have been described in practice, it is unlikely that the transported AI doses would be continuously exposed to strong vibrations, as demonstrated by our experiment. Nevertheless, if it is assumed that the effects of short, violent vibrations as well as long, milder vibrations will accumulate during a shipping tour, then our findings remain a valid reflection of these conditions. Regardless, this cumulative effect has yet to be demonstrated with a specific experimental design.

## 5. Conclusions

Liquid-preserved boar semen is a perishable product and should be transported carefully. As not only the intensity but also the duration of vibration emissions is decisive for the influence on sperm quality, a shipping route to the customer with as little vibration as possible should be selected, especially for longer transport times and BTS-extended semen. If there is no alternative to a route with rough road conditions, the driving behavior should be adjusted by reducing speed [10]. A real-time monitoring system could evaluate the success of these and other compensatory measures.

## Figures and Tables

**Figure 1 animals-13-00952-f001:**
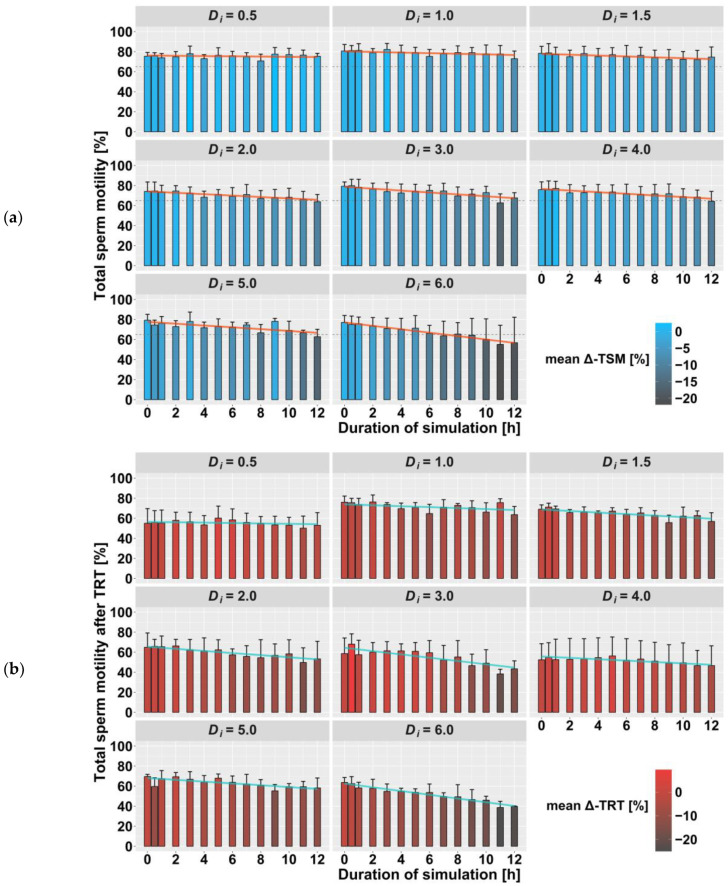
Summary of CASA measurements on day 4 after treatment: (**a**) total sperm motility after 10 min incubation at 38 °C (TSM, *n* = 546 samples); (**b**) total sperm motility after 300 min incubation at 38 °C (thermo-resistance test TRT, *n* = 336 samples). Mean absolute values, grouped by intensity (D_i_) and duration (h) of vibration treatment, are shown as bars with standard deviation (SD) as the error bar. The color of the bars represents the mean difference (Δ) to the control group (duration = 0 h). The colored line shows the values of the applied regression model, and the dashed line in (**a**) stands for the minimum requirement of 65% motile spermatozoa on the expiration day.

**Figure 2 animals-13-00952-f002:**
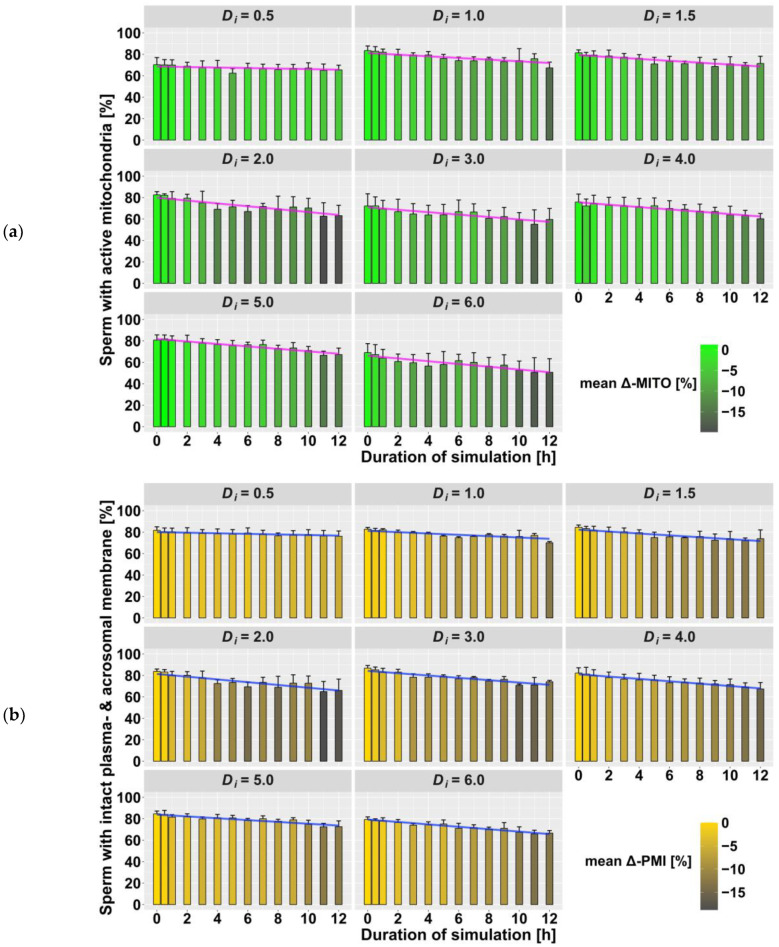
Summary of flow cytometric measurements (*n* = 336 samples) on day 4 after treatment: (**a**) mitochondrial activity (MITO); (**b**) plasma membrane and acrosome integrity (PMI). Mean absolute values, grouped by intensity (D_i_) and duration (h) of vibration treatment, are shown as bars with standard deviation (SD) as the error bar. The color of the bars represents the mean difference (Δ) to the control group (duration = 0 h). The colored line shows the values of the applied regression model.

**Table 1 animals-13-00952-t001:** Experimental design: every examined ejaculate was split into 14 subsamples (*n* = 546 in total) and treated for 13 different durations (0.5 to 12 h) with rising vibration intensities (D_i_). The unshaken sample served as a control (duration = 0 h).

Displacement Index (D_i_)	Corresponding Road Type ^1^	Ejaculate Numbers (n) Examined with CASA
0.5	Smooth asphalt	3
1.0		3
1.5	Rough asphalt	6
2.0		6
3.0	Cobblestone	6
4.0		6
5.0		3
6.0		6

^1^ According to Hafemeister et al. [10].

**Table 2 animals-13-00952-t002:** Summarized impact of different vibration intensities (D_i_ ± 0.1) on different sperm quality characteristics on d4 of semen storage, representing short (1 h), middle (6 h) and long (12 h) transport simulation.

Displacement Index (D_i_)	Duration (h)	Δ-TSM (%)	Δ-TRT (%)	Δ-MITO (%)	Δ-PMI (%)
Control (n = 39, absolute values)	0 (non-shaken)	77.33 ± 6.59	63.73 ± 7.96	76.99 ± 5.83	83.25 ± 2.23
0.5	1	−1.5 ± 0.3	0.2 ± 1.2	−0.4 ± 1.1	−1.4 ± 0.5
	6	0.4 ± 1.3	3.4 ± 2.2	−2.9 ± 2	−2.4 ± 1
	12	0 ± 0.7	−2.1 ± 1.2	−4.9 ± 1.3	−5.5 ± 0.9
1.0	1	0.5 ± 0.2	−2.6 ± 0.9	−1.5 ± 0.8	−0.6 ± 1.4
	6	−5.4 ± 0.7	−11.4 ± 3.9	−9.4 ± 0.8	−8.1 ± 1.2
	12	−7.8 ± 1	−12.4 ± 2.7	−16.2 ± 0.9	−12.5 ± 1.1
1.5	1	−1.1 ± 2.2	0.3 ± 1.2	−2.1 ± 0.7	−2.7 ± 0.7
	6	−3.4 ± 2.7	−5 ± 0.8	−7.9 ± 2	−8.9 ± 1.3
	12	−3.8 ± 2.7	−12.2 ± 4.3	−9.9 ± 2.9	−10.5 ± 2.6
2.0	1	−0.8 ± 1.6	0.6 ± 2.5	−3.3 ± 1.4	−3.9 ± 0.7
	6	−5 ± 1.9	−7.5 ± 3.7	−15.5 ± 1.1	−14.3 ± 0.9
	12	−10.2 ± 2.8	−11.6 ± 2.3	−19.4 ± 3.2	−17.7 ± 3.7
3.0	1	−1 ± 2.1	−1.2 ± 5.1	−2.7 ± 1.6	−3.8 ± 0.8
	6	−4.2 ± 1.5	0.8 ± 2.2	−5.2 ± 0.6	−9.9 ± 0.4
	12	−11.7 ± 2.5	−15.3 ± 5.5	−12.5 ± 0.7	−12.6 ± 1.5
4.0	1	1.2 ± 1.3	0.1 ± 1.9	−1.4 ± 0.4	−2.6 ± 0.4
	6	−3.9 ± 1.5	−0.5 ± 3.3	−6 ± 1.1	−9.1 ± 0.7
	12	−11.6 ± 1.9	−5.7 ± 3.6	−15.6 ± 1.6	−15 ± 2.3
5.0	1	−3.2 ± 0.6	−2.3 ± 4.9	0 ± 0.6	−3.1 ± 1.6
	6	−6.9 ± 2.4	−5.9 ± 3.5	−4.4 ± 1.5	−6 ± 0.5
	12	−16.4 ± 7.1	−11.4 ± 6.1	−13.6 ± 5.2	−11.9 ± 3.9
6.0	1	−1.3 ± 2.3	−5.5 ± 0.7	−5 ± 1.4	−1.4 ± 0.3
	6	−10.9 ± 2.9	−10.1 ± 4.1	−7.5 ± 1.2	−8.3 ± 1.6
	12	−20.3 ± 8.9	−23.9 ± 2.1	−18.3 ± 3.9	−12.8 ± 0.8

TSM—total sperm motility, TRT—thermo-resistance test, MITO—spermatozoa with active mitochondria, PMI—spermatozoa with intact plasma- and acrosomal membranes; Δ: difference between treatment and control of the same ejaculate; values are presented as mean ± standard error.

**Table 3 animals-13-00952-t003:** The results of the regression analysis are shown, explaining the impact of vibrations on sperm quality characteristics during transport. The linear mixed model includes the interaction of vibration intensity (D_i_) and duration of simulation as fixed effects and the boar as a random effect on intercept and slope. In addition, the storage days (d1–4) were considered for total sperm motility.

Dependent Variable	Independent Variable	Estimates	SE	df	t-Value	*p*-Value
Total sperm motility	*(Intercept)*	79.79	0.99	41	80.42	
	Storage day	−0.66	0.08	2104	−8.10	<0.001
	D_i_ × duration	−0.30	0.03	32	−11.75	<0.001
Thermo-resistance test	*(Intercept)*	64.46	2.08	23	30.98	
	D_i_ × duration	−0.39	0.06	16	−6.95	<0.001
Mitochondrial activity	*(Intercept)*	75.48	1.62	23	46.74	
	D_i_ × duration	−0.45	0.06	17	−7.59	<0.001
Plasma membrane integrity	*(Intercept)*	81.74	0.66	23	123.41	
	D_i_ × duration	−0.43	0.05	20	−7.93	<0.001

D_i_ × duration—interaction between displacement index and duration, SE—standard error, df—degrees of freedom.

## Data Availability

The data presented in this study are available on request from the corresponding author.

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
