# Peer review of "Intensity and Duration of Vibration Emissions during Shipping as Interacting Factors on the Quality of Boar Semen Extended in Beltsville Thawing Solution"

_animals, 2023, doi:10.3390/ani13050952_

Round 1
Reviewer 1 Report
This manuscript describes quality metrics (motility, thermos-resistance, mitochondrial activity, membrane integrity) of extended boar semen that was subject to simulated vibration emissions over varying time periods (i.e. duration of transport). These emissions were meant to simulate the effects of ground transport over different surfaces on quality of boar semen intended for use in AI.
Line 46
What is the average transport time of commercial AI doses among these facilities? Are most traveling for 3-4 days over rough terrain, or is that a minimal number?
Line 60-61
Additional details about the sensing app and how it works would be helpful. Citation 9 seems to be in German, so most readers won’t be able to review this background on their own. Brief mention of what was concluded from reference 7 would also be helpful (a sentence would suffice)
Line 87
Is Beltsville Thawing Solution extender typically used by the majority of swine producers in Europe? Are other extenders used? If so, at what frequency? If other extenders are more commonly, or just as commonly, used as the BTS that should be noted. The potential influence of extender is touched on in the conclusions, but should be mentioned in the intro
Author Response
Referee #1
This manuscript describes quality metrics (motility, thermo-resistance, mitochondrial activity, membrane integrity) of extended boar semen that was subject to simulated vibration emissions over varying time periods (i.e. duration of transport). These emissions were meant to simulate the effects of ground transport over different surfaces on quality of boar semen intended for use in AI.
AC: First of all, we would like to thank you very much for taking the time to review our manuscript. Your comments were of high value to us and we hope, that we answered your questions sufficiently. The entire manuscript has been critically revised.
Line 46
What is the average transport time of commercial AI doses among these facilities?
Are most traveling for 3-4 days over rough terrain, or is that a minimal number?
AC: Thank you for this important remark. Relevant information has been added to the manuscript.
Line 60-61
Additional details about the sensing app and how it works would be helpful.
Citation 9 seems to be in German, so most readers won’t be able to review this background on their own. Brief mention of what was concluded from reference 7 would also be helpful (a sentence would suffice).
AC: Thank you very much for pointing this out. Although it is described more in detail in reference 9, the operation of the mobile sensing app is also explained in reference 7 in English. The functionality of the mobile sensing app is described in detail in Reference 9. There is also an abstract in English for this. We have added reference 7 to reference 9. There, too, the background of the app is explained in detail.
Reference 7 was able to show the effect of vibration as a function of intensity. However, this study considered only one transport duration (= 6 hours). Mentioning this fact illustrates the usefulness of the current study, in which transport times of 30 min up to 12 hours are simulated.
Line 87
Is Beltsville Thawing Solution extender typically used by the majority of swine producers in Europe? Are other extenders used? If so, at what frequency? If other extenders are more commonly, or just as commonly, used as the BTS that should be noted. The potential influence of extender is touched on in the conclusions, but should be mentioned in the intro
AC: Thank you for this important comment. Although long-term extenders are used more frequently, the very simple and above all inexpensive BTS extender dominates in Europe and especially in Germany (about 75% market volume). Of course, the use of long-term extenders can significantly reduce the influence of vibrations on sperm quality. This has already been shown in previous studies and is also reported in the introduction.
Reviewer 2 Report
Dear Authors,
In my opinion, the presented research may have some practical significance. Nevertheless, it is well known that the storage time and the increase in vibration intensity adversely affect semen quality. It is a pity that the authors failed to determine the threshold values for vibration at which there is a significant deterioration in the quality of preserved sperm. In connection with the researched factor, I have a question and a suggestion. Can the applied vibration values be related to the vibrations that occur on the roads on which the material is transported? It is worth including such information in the discussion. Have the vibration values on the roads been tested?
In general, the manuscript is prepared correctly. I have comments regarding the presentation of Figures 1 and 2. Marking statistical differences with color is not very legible. I suggest to mark these differences with letters, for example.
Author Response
Referee #2
Dear Authors, in my opinion, the presented research may have some practical significance. Nevertheless, it is well known that the storage time and the increase in vibration intensity adversely affect semen quality.
AC: Thank you for reviewing our work. We appreciate your insight and hope that our comments and the revised manuscript meet your expectations.
Unfortunately, the study situation on sperm-damaging vibration is not so clear, and especially the interaction of the factors intensity and duration has not been analyzed yet. You are correct that storage duration is known to affect sperm quality. However, since this was affected by our study design, we had to take it into account in the analysis of the data.
It is a pity that the authors failed to determine the threshold values for vibration at which there is a significant deterioration in the quality of preserved sperm.
AC: Thank you very much for pointing this out. However, due to the very heterogeneous baseline quality of ejaculates, it is not possible to determine the damage to spermatozoa using a single cut-off value. We were able to show for the first time that sperm quality steadily decreases as a function of intensity and duration (interaction term). Ultimately, it depends on the initial quality of the ejaculates how much of vibration leads to insufficient sperm quality and how we define good and poor sperm quality. Unfortunately, the cut-off values for ejaculates suitable for use in artificial insemination (AI practice) are not uniform worldwide (reviewed in Waberski D, Riesenbeck A, Schulze M, Weitze KF, Johnson L. Application of preserved boar semen for artificial insemination: Past, present and future challenges. Theriogenology, 2019, 137:2-7).
In connection with the researched factor, I have a question and a suggestion. Can the applied vibration values be related to the vibrations that occur on the roads on which the material is transported? It is worth including such information in the discussion. Have the vibration values on the roads been tested?
AC: Thank you for the good point. The preliminary study detailing the vibrations experienced during real boar semen deliveries in Germany, Brazil and USA has been published and cited (Reference 11), and the relationship between the values and road types is summarized in Table 1. In addition, the simulation of the extreme vibration intensities is discussed in lines 302 to 310.
In general, the manuscript is prepared correctly. I have comments regarding the presentation of Figures 1 and 2. Marking statistical differences with color is not very legible. I suggest to mark these differences with letters, for example.
AC: Thank you very much for this valuable reference. In the figures, several independent types of information are shown simultaneously. The color coding in the figures only shows the calculated difference in the quality parameter of the respective treatment group compared to the unshaken control group. Thus, the initial ejaculate quality was also taken into account. The statistical classification is only done by the regression line, which represents the statistical model. Its meaning is described in the text and in Table 3.
Reviewer 3 Report
Well done on a well-designed and -executed study, with robust results of direct application to the swine industry.
I have a few suggestions regarding language and content:
Line 12, 42, 311: Replace ‘vulnerable’ with ‘perishable’
67: Clarify the intended meaning of the phrase ‘in a wide range’
69 and 212: Delete ‘emissions’
88: Replace ‘filled’ with ‘placed’ (or ‘decanted into’)
90: Rephrase: ‘Compliance with the minimum requirements ….’
165-166: It is not clear how examination day reflects the influence of transport, rather than storage time.
227: ‘revolutions per minute, rpm’
301: Replace ‘in’ with ‘across’, and add ‘…. of vibration intensity and duration’ for clarity
338: Contrary to the statement of no conflicts of interest, one author has a commercial interest in the semen extender and containers used in this project, constituting a COI which should be acknowledged.
The authors could consider proposing an equation based on their results to calculate a vibration quantum which considers both duration and intensity of vibrations. Such an equation would be useful for shippers to use as a decision-making tool in considering how semen samples could be moved to their destination while still maintaining their quality above the minimum standards.
Author Response
Referee #3
Well done on a well-designed and -executed study, with robust results of direct application to the swine industry.
AC: First of all, we would like to thank you very much for taking the time to review our manuscript. Your comments were of high value to us and we hope, that we answered your questions sufficiently. The entire manuscript has been critically revised.
I have a few suggestions regarding language and content:
Line 12, 42, 311: Replace ‘vulnerable’ with ‘perishable’
88: Replace ‘filled’ with ‘placed’ (or ‘decanted into’)
90: Rephrase: ‘Compliance with the minimum requirements ….’
227: ‘revolutions per minute, rpm’
301: Replace ‘in’ with ‘across’, and add ‘…. of vibration intensity and duration’ for clarity
AC: Thank you for your recommendations. They have been implemented.
67: Clarify the intended meaning of the phrase ‘in a wide range’
AC: The corresponding phrase was explained in detail and simulated transport conditions were added to the manuscript.
69 and 212: Delete ‘emissions’
AC: The technical term "emissions" means that the vibrations act directly on the sperm. Where possible, we have reduced the term to "vibrations".
165-166: It is not clear how examination day reflects the influence of transport, rather than storage time.
AC: We hope the present explanation makes it clearer.
338: Contrary to the statement of no conflicts of interest, one author has a commercial interest in the semen extender and containers used in this project, constituting a COI which should be acknowledged.
AC: The author C. Simmet actively participated as a scientist in this project and is therefore also a co-author of the study. The study was supported by funds from the Federal Ministry of Economy and Climate Action in Germany and not by the company Minitüb. Appropriate information about our project IQTrans can be read on our homepage (https://www.iqtrans-projekt.de/). Data on the project must be published and disclosed. This has been done herewith. Also the results of the study do not bring any economic advantage to the company Minitüb, because the used extender (BTS) was not able to neutralize the simulated vibrations. Accordingly, there is no COI to declare. This has already been reported to the Editorial office.
The authors could consider proposing an equation based on their results to calculate a vibration quantum which considers both duration and intensity of vibrations. Such an equation would be useful for shippers to use as a decision-making tool in considering how semen samples could be moved to their destination while still maintaining their quality above the minimum standards.
AC: Internally, we have also discussed the possibility of specifying an equation. However, we came to the conclusion that this would not provide any relevant additional information. All necessary parts for the calculation of the average sperm quality loss can be found in this paper or in the preliminary study [11].